

# Knowledge and attitudes toward pediatric pain management among nursing interns from a selected university in Riyadh, Saudi Arabia: a descriptive cross-sectional quantitative study

Sumathi Robert Shanmugam[1], Amany Anwar Saeed Alabdullah[1], Maha Hanis Alenezi[2], Shahad Khalid Aldughyshim[2], Maryam Fahad Alnemer[2], Wedad Khalid Almutairi[2], Ghaida Saad Alhadyan[2], Rasha Zaid Albugomi[2] and Fatimah Abdullah Alkhulayfi[2]

[1] Department of Maternity and Pediatric Nursing, College of Nursing, Princess Nourah bint Abdulrahman University, Riyadh, Saudi Arabia
[2] College of Nursing, Princess Nourah bint Abdulrahman University, Riyadh, Saudi Arabia

Corresponding author
Amany Anwar Saeed Alabdullah,
aaalabdullah@pnu.edu.sa

## ABSTRACT

**Introduction**. Pediatric pain is often not addressed properly in the literature, which suggests a research gap in pediatric health-care providers' knowledge and attitudes toward the treatment of pain experienced by children in various health-care settings. To improve future practice in this area, nursing interns should be well versed in pediatric pain assessment and management to improve pediatric pain management practices in collaboration with other health-care professionals.

**Purpose**. This study aimed to assess the levels of knowledge and attitudes toward pediatric pain management among nursing interns at a specific academic institution.

**Methods**. This descriptive cross-sectional quantitative study employed an online questionnaire to collect data from 119 female nursing interns in Riyadh, Saudi Arabia. In addition to collecting the participants' demographic profiles, the questionnaire gathered data using the Pediatric Nurses' Knowledge and Attitudes Survey Regarding Pain instrument. Descriptive and inferential statistics were calculated using SPSS for Windows (v. 21.0).

**Results**. The nursing interns' overall knowledge and attitudes toward pediatric pain management were found to be poor, with a mean score of 36.59% (standard deviation, 13.2).

**Conclusion**. Additional education and clinical training for nursing interns is essential to enhance their knowledge and attitudes toward pediatric pain management.

# INTRODUCTION

As one of the key symptoms of many health conditions, pain can be experienced for a variety of etiological reasons. Pain negatively affects individuals' social life, physical and

mental health, and overall quality of life. It is a considerable source of distress for children and their families, as well as health-care workers. Surgery, trauma, acute and chronic illnesses, and medical or surgical treatments can all cause pain. Although pain affects individuals of all ages, it can have long-term physiological, psychosocial, and behavioral effects on children if untreated (*Gadallah, Hassan & Shargawy, 2017*).

*Dezfouli & Khosravi (2020)* observed that treating pain in children can be challenging, considering the need to minimize opioid use to avoid their adverse effects. However, a variety of pharmacologic and nonpharmacologic pain management options are available for children who experience acute and chronic pain. For example, nociceptive and neuropathic pain can be targeted by opioids and opioid-sparing medicines. Thus, the early use of a combination of pharmacological and integrative nonpharmacological pain management options is crucial for the treatment of pediatric pain.

Nurses are in a unique position to effectively diagnose and manage pediatric pain because they constitute the majority of health-care workers and spend the most time interacting with children and their families during hospitalization (*Alotaibi, 2019*). However, unresolved pediatric pain remains an issue due to a lack of information and prevailing attitudes toward its management among most health-care professionals, including nurses. Therefore, nurses and other health-care workers must have sufficient information and positive attitudes to enhance the assessment and management of pediatric pain. As core members of health-care teams, nurses play a vital role in the assessment and management of pain.

Pediatric nursing is included in the undergraduate nursing curriculum as a whole, which makes learning about pain management difficult. *Mediani, Windiany & Maryam (2020)* found that undergraduate nursing students have insufficient knowledge about and poor attitudes toward pain assessment and treatment in children, which influences their provision of pediatric nursing care. In providing improved pain management outcomes for their pediatric patients, nurses should ideally become change agents in their hospitals. If nursing students are not well versed in pediatric pain and its management, they will not be able to appropriately relieve the pain experienced by their pediatric patients. Thus, nursing students should have adequate knowledge about pain assessment and management in pediatric patients and should endeavor to improve their pain management practices by collaborating with other health-care professionals. Furthermore, nursing students must understand how to develop positive attitudes toward pain management and apply their knowledge in a clinical setting while studying in their undergraduate nursing programs. Pediatric patients experiencing pain will suffer if nursing students lack the necessary information and training to adequately assess and manage pediatric pain. Therefore, nursing students must gain a thorough understanding of pain and how to manage it while completing their undergraduate nursing programs.

## Objective

The objective of this study was to assess the levels of nursing interns' knowledge about and attitudes toward pediatric pain management at a specific academic institution.

### Research questions

This study explored the following two research questions: (1) What are the nursing interns' demographic profiles in terms of age, marital status, grade point average (GPA), training hospital, and area of training? (2) What are the nursing interns' current attitudes and levels of knowledge regarding pediatric pain management?

## MATERIALS AND METHODS

### Study design and setting

This study adopted a descriptive cross-sectional quantitative design to assess the knowledge and attitudes regarding pediatric pain management among female nursing interns at a specific academic institution to which only female students are admitted.

### Participants

The study population comprised 171 female nursing interns who were placed in eight hospitals in the Riyadh region of Saudi Arabia. Using an app to calculate the estimated proportional impact, a minimum sample size of 119 was deemed necessary to have a 95% confidence interval and a 5% margin of error (MOE) to ensure that the true proportion of nursing interns in the entire population fell within the established interval calculated from the sample. The following EPI Apps (*Fellows Statistics, 2025*) formula was used to calculate the sample size: $n = N \times X / (X + N - 1)$, where $N$ is the population size (*i.e.,* 171), and when confidence is 95%, $X = (1.96)^2 \times p \times (1 - p)/\text{MOE}^2$ (MOE = 0.05 in this case; for the purposes of the sample calculation, the sample proportion, $p$, was set to 0.5).

### Instrument

The nursing interns' demographic information, such as their age, marital status, GPA, and the name of their training hospital, was collected. Following a literature review, a survey with established validity and reliability, the Pediatric Nurses' Knowledge and Attitudes Survey Regarding Pain (PNKAS) (*Manworren, 2000*), was selected as the most appropriate tool. Permission to use the PNKAS was obtained from the copyright holders.

*Manworren (2000)* examined the reliability and validity of the PNKAS instrument, and its face and content validity were determined using a panel of five nursing specialists in pain management. The test–retest reliability correlation coefficient was 0.67 among six nurses and six child life specialists, indicating a high level of instrument stability. Cronbach's $\alpha$ for the PNKAS instrument was calculated using data from two groups of pediatric nurses and varied from 0.72 to 0.77, indicating that the instrument's level of internal consistency was satisfactory. Eight pediatric experts (seven registered nurses and one pediatrician) confirmed the instrument's face validity.

The PNKAS instrument includes 22 true-or-false questions, 16 multiple-choice questions, and two case studies, which are explored through four multiple-choice questions. The PNKAS takes approximately 30–40 min to complete. A correctly answered question was scored as one point, whereas an incorrectly answered question was scored as zero points. Hence, the range of scores for the 42-item PNKAS was 0–42. The total scores were converted into percentages for each participant using the following formula: total

**Table 1  Frequency and percentage distribution of the nursing students according to their demographic profiles.**

| Profile | Frequency | Percentage |
|---|---|---|
| **Age (years)** | | |
| 18–20 | 2 | 1.7 |
| 21–23 | 109 | 91.6 |
| 24–25 | 8 | 8.6 |
| **Marital status** | | |
| Single | 109 | 91.6 |
| Married | 10 | 8.4 |
| **GPA** | | |
| 2.00–2.74 | 1 | 0.8 |
| 2.75–3.74 | 6 | 5.0 |
| 3.75–4.49 | 83 | 69.7 |
| 4.50–5.00 | 29 | 24.4 |
| **Clinical area** | | |
| Pediatric unit | 23 | 19.3 |
| Non-pediatric unit | 96 | 80.7 |

percentage score = (total score obtained/42) ×100. The resulting scores were interpreted as follows: good, 75–100%; average, 51–74%; poor, ≤50%.

## Data collection

The data were collected from February to March 2022 using a Google Forms questionnaire. Nursing intern representatives were contacted *via* WhatsApp, and the survey link was distributed by email to the nursing interns.

## Statistical analysis

The data were coded and analyzed using the Statistical Package for the Social Sciences (SPSS) for Windows (v. 21.0; IBM SPSS, Armonk, NY, USA). Tables 1–4 and a bar chart (Fig. 1) were developed to organize the data collected from each question, including descriptive statistics for frequency, percentage, mean, and standard deviation (SD). The $r$ value was used as a measure of association. The frequency and percentage distributions were used to summarize and organize the respondents' demographic profiles, academic performance, areas of work, and levels of knowledge and attitudes toward pediatric pain management. No missing data and no questions received more than one response from any of the participants.

## Ethical considerations

Institutional review board approval was obtained from an academic institution in Riyadh, Saudi Arabia (approval number: 22-0110). The confidentiality and privacy of the participants' data were guaranteed, and a written consent was obtained before data collection began. At the start of the questionnaire, participants answered a question indicating that the completion and submission of the questionnaire *via* Google Forms served as implied consent.

**Table 2** Frequency and percentage distribution of knowledge and attitudes toward pediatric pain management (mean = 36.59, SD = 13.02).

| Grade | Performance level | Frequency | Percentage |
|---|---|---|---|
| 75–100 | Good | 3 | 2.5 |
| 51–74 | Average | 13 | 10.9 |
| ≤50 | Poor | 103 | 86.5 |

**Table 3** Top 10 correctly answered test items.

| Knowledge and attitude test items | Freq | % age |
|---|---|---|
| 1. *Item 22.* After the initial recommended dose of opioid analgesic, subsequent doses should be adjusted in accordance with the individual patient's response. | 96 | 80.7 |
| 2. *Item 8.* Children who require repeated painful procedures (eg, daily blood draws) should receive maximum treatment for the pain and anxiety of the first procedure to minimize the development of anticipatory anxiety before subsequent procedures. | 89 | 74.8 |
| 2. *Item 11.* Combining analgesics (eg, acetaminophen and topical anesthetics) with non-drug therapies (eg, sucrose and nonnutritive sucking) that work via different mechanisms may result in better pain control with fewer side effects than using a single analgesic agent. | 89 | 74.8 |
| 4. *Item 30.* Analgesics for postoperative pain should be given initially. | 80 | 67.2 |
| 5. *Item 5.* Comparable stimuli produce the same intensity of pain in different people. | 73 | 61.3 |
| 6. *Item 20.* Based on one's religious beliefs, a child/adolescent may think that pain and suffering are necessary. | 69 | 58.0 |
| 7. *Item 32.* Analgesia should be given for chronic cancer pain. | 67 | 56.3 |
| 8. *Item 14.* Parents should not be present during painful procedures. | 65 | 54.6 |
| 8. *Item 18.* A child/adolescent with pain should be encouraged to endure as much pain as possible before resorting to a pain relief measure. | 65 | 54.6 |
| 10. *Item 10.* Acetaminophen 650 mg orally is approximately equal to codeine 32 mg orally in analgesic effect. | 64 | 53.8 |

## RESULTS

Of the 119 nursing interns who participated in the survey, the majority (109, 91.6%) were aged 21–23 years, 24–25 years (8, 8.6%), and 18–20 years (2, 1.7%). Almost all participants (109, 91.6%) were single, and only 10 (8.4%) were married. The largest proportion of participants (83, 69.7%) had a GPA of 3.75–4.49, followed by 29 (24.4%) with a GPA of 4.50–5.00, and 6 (5%) with a GPA of 2.75–3.74. Only 1 (0.8%) had a GPA of 2.00–2.74. Approximately 23 (19.3%) of the respondents were assigned as interns in pediatric units,

**Table 4  Top 10 incorrectly answered test items.**

| Knowledge and attitude test items | Freq | % age |
|---|---|---|
| 1. *Item 26.* The recommended route of administration of opioid analgesics to children with prolonged cancer-related pain. | 108 | 91.0 |
| 2. *Item 40.* Your assessment above was made 2 h after he received morphine 2 mg IV. After he received the morphine, his pain ratings every half hour ranged from 6 to 8, and he had no clinically significant respiratory depression, sedation, or other untoward side effects. He identified 2 as an acceptable level of pain. His physician's order for analgesia was ''morphine IV 1–3 mg 1 h PRN pain relief.'' Check the action you would take at this time. | 105 | 88.2 |
| 3. *Item 7.* Evidence-based non-drug interventions are effective for mild–moderate pain control but are rarely helpful for more severe pain. | 96 | 80.7 |
| 4. *Item 1.* Observable changes in vital signs must be relied upon to verify a child's/adolescent's statement that they have severe pain. | 95 | 79.8 |
| 4. *Item 15.* Adolescents with a history of substance abuse should not be given opioids for pain because they are at a high risk of addiction loop. | 95 | 79.8 |
| 6. *Item 42.* Your assessment above was made 2 h after he received morphine 2 mg IV. After he received morphine, his pain ratings every half hour ranged from 6 to 8, and he had no clinically significant respiratory depression, sedation, or other untoward side effects. He identified 2 as an acceptable level of pain. His physician's order for analgesia was ''morphine IV 1–3 mg 1 h PRN pain relief.'' Check the action you would take at this time. | 93 | 78.2 |
| 7. *Item 31.* A child with chronic cancer pain has been receiving daily opioid analgesics for 2 months. The doses gradually increased during this period. Yesterday, the child received morphine 20 mg/h intravenously. Today, he has been receiving 25 mg/hour intravenously for 3 h. The likelihood that the child will develop clinically significant respiratory depression is as follows: | 91 | 76.5 |
| 7. *Item 36.* Which of the following describes the best approach for cultural considerations in caring for a child/adolescent in pain? | 91 | 76.5 |
| 9. *Item 25.* For effectiveness, heat and cold should be applied directly to painful areas. | 87 | 73.1 |
| 10. *Item 17.* Young infants less than 6 months of age cannot tolerate opioids for pain relief. | 82 | 68.9 |

while the majority (96, 80.7%) were assigned as interns in various nonpediatric areas (Table 1).

## Knowledge and attitudes toward pediatric pain management

Table 2 shows the results for the nursing interns' levels of knowledge and attitudes regarding pediatric pain management. Among all students who participated in the survey, only 3 (2.5%) had a good level (75–100%) of knowledge and positive attitudes toward pediatric

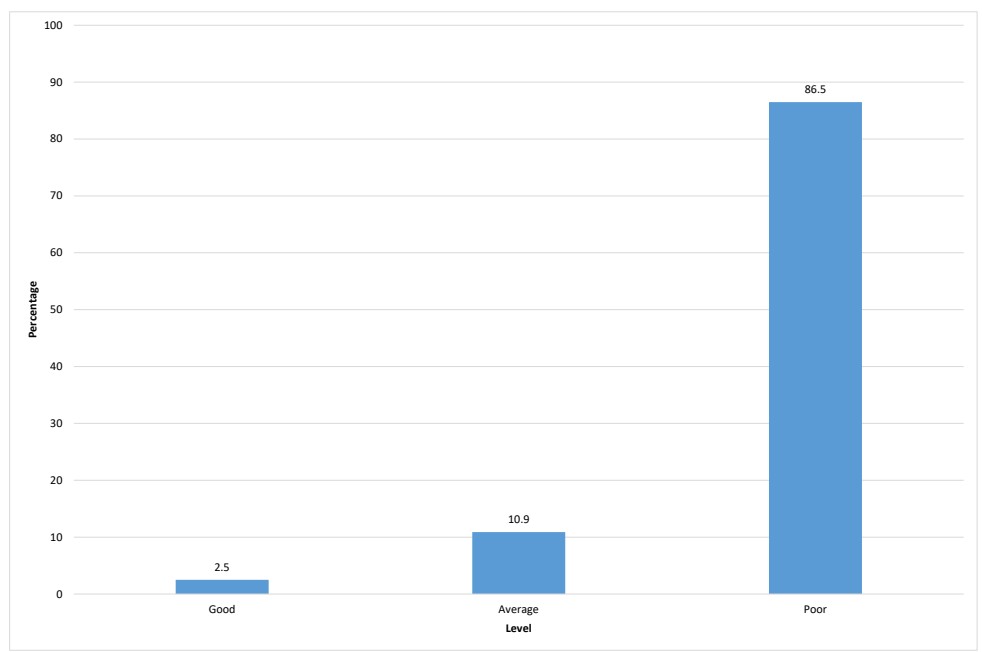

**Figure 1  Knowledge and attitudes toward pediatric pain management.**

pain management, while the majority (103, 86.5%) had poor levels (≤50%) and 13 (10.9%) had average levels (51–74%).

Figure 1 depicts the distribution of the levels of knowledge and attitudes toward pediatric pain management. The data were not normally distributed but skewed to the right (toward the lower end of the scale), with a high percentage of students with poor levels of knowledge and attitudes toward pediatric pain management, while the percentage of students with good scores was extremely small.

The top 10 correctly answered questions from both the true-or-false and multiple-choice sections of the questionnaire were selected and analyzed to determine the areas that were best understood by the participants (Table 3). These items were presented with the highest to lowest frequency of correct answers, ranging from 80.7% to 53.8% of the respondents.

The 10 items from the true-or-false and multiple-choice questions on pediatric pain management that were most frequently answered incorrectly by the nursing interns were selected to determine the aspects of pain management for which their knowledge was most lacking (Table 4). These items are presented in the order of the frequency of incorrect answers, ranging from 91% to 68.9% of respondents who answered incorrectly.

## DISCUSSION

The present study aimed to assess nursing interns' levels of knowledge and attitudes toward pediatric pain management at an academic institution. All participants were female nursing interns, unlike the earlier studies by *Al Omari (2016)* and *Alotaibi (2019)*. To the best of our

knowledge, our study is the first to include only female nursing interns from an academic institution in Riyadh, Saudi Arabia.

The current study revealed that the nursing interns' overall knowledge and attitudes with respect to pediatric pain management were poor, with a mean score of 36.59 (SD, 13.02). *Shdaifat, Al-Shdayfat & Sudqi (2020)* used the same instrument among nursing students and recorded higher scores on average (mean, 42.6; SD, 9.1). This finding is congruent with previous studies (*Chow & Chan, 2015*; *Ortiz et al., 2015*; *Al Omari, 2016*; *Laprise, 2016*; *Gadallah, Hassan & Shargawy, 2017*; *Kusi Amponsah et al., 2020*; *Mediani, Windiany & Maryam, 2020*; *Shdaifat, Al-Shdayfat & Sudqi, 2020*). An exception was a study by *Aydın & Bektaş (2020)* who administered a pain management knowledge questionnaire among nursing interns and found a moderate level of knowledge.

A recent study in Saudi Arabia (*Alotaibi, 2019*) also had comparable results. Based on the findings of this study and others, nursing students currently appear to be insufficiently prepared for pediatric pain management. Moreover, the little time spent with children in the clinical field and the limited opportunities provided were reasons suggested for this study's results.

The most correctly answered items in this study were similar to those obtained by *Gadallah, Hassan & Shargawy (2017)*, *Alotaibi (2019)*, *Kusi Amponsah et al. (2020)*, and *Shdaifat, Al-Shdayfat & Sudqi (2020)*. The items with correct responses about pain assessment and individualized pain experiences and their treatments may be due to the simplicity of the questions, which made them more comprehensible to the nursing interns.

Insufficient knowledge of pain assessment was evident in the nursing interns' incorrect responses. More than two-thirds of the nursing interns' responses were incorrect for the item about changes in vital signs as an indicator of severe pain. The nursing interns did not understand that children present changes in vital signs as a physiological response to pain. This finding was also evident in the studies by *Chiang et al. (2006)*, *Al Omari (2016)* and *Gadallah, Hassan & Shargawy (2017)*. This finding indicates that nursing students must be taught appropriate pediatric pain assessment.

The study results imply that nursing interns' responses regarding pharmacological management were mostly incorrect. For example, most of the study participants showed extremely poor knowledge of the route of administration of opioid analgesics and pharmacological items related to the two case studies. This result echoes the studies by *Chow & Chan (2015)*, *Al Omari (2016)*, and *Alotaibi (2019)*, which showed a similar percentage of responses.

It was also evident from this study that the nursing interns exhibited poor knowledge about nonpharmacological approaches to pediatric pain management, as represented by their incorrect responses to items 7 and 25. Similarly, *Chiang et al. (2006)* and *Mediani, Windiany & Maryam (2020)* found that a considerable number of nursing students provided incorrect responses in this area.

The study results indicated no significant differences in the nursing interns' knowledge and attitude scores with regard to their demographic data. While this finding is similar to the findings of *Karaman, Vural Doğru & Yıldırım (2018)*, it contradicts a study by *Alotaibi (2019)* conducted in Saudi Arabia, which found that nursing students' scores

showed substantial differences according to their gender, age, clinical experience, and qualifications.

For several of these results, multiple factors may explain the nursing interns' or students' poor knowledge and attitudes toward pediatric pain assessment and its management. For example, the nursing curriculum may lack conceptual content describing the assessment and management of pediatric pain, and nursing students may be provided with less clinical exposure to pediatric care.

### Limitations

One of the limitations of this study is that the results cannot be applied to the entire population due to the small sample size and all the participants were female. In terms of the clinical area of assignment/training in the hospital, most of the nursing interns in this study were assigned to nonpediatric areas, with only a few working in pediatric areas. Moreover, the nursing curriculum's provides a broad overview of pediatric nursing care, with relatively less emphasis placed on pain management, which may have impacted the interns' knowledge and experience in this area.

Another limitation was the limited time available for data collection, as the nursing interns were frequently busy, making it difficult to obtain responses and administer the instrument within the short periods of their availability.

## CONCLUSION

Despite pain management in children being a vital aspect of pediatric practice, this study demonstrated nursing interns' poor knowledge and attitudes toward pediatric pain management, especially pharmacological and nonpharmacological pain management. This finding could be the result of a lack of details included in the nursing school curriculum and their short clinical training period as students in pediatric care in hospitals, which was mostly observational rather than practical.

### Recommendations

This study provides insights into the importance of redesigning nursing curricula with clinical training to strengthen nursing students' knowledge of pediatric pain assessment. Teaching and training methods must be revised to help nursing students practice their skills in pediatric pain assessment and management, with close supervision by clinical nursing educators. These findings should be disseminated to hospital administrators, academicians, and policymakers to help initiate programs to improve skills in pediatric pain management. In-service education may be tailored to nurses and interns to update their knowledge of pediatric pain management practices. Further research studies are recommended to focus on nurses' and nursing students' knowledge and attitudes in different settings.

### Funding

This research was funded by Princess Nourah bint Abdulrahman University Researchers Supporting Project number (PNURSP2025R444), Princess Nourah bint Abdulrahman University, Riyadh, Saudi Arabia. The funders had no role in study design, data collection and analysis, decision to publish, or preparation of the manuscript.

### Grant Disclosures

The following grant information was disclosed by the authors:
Princess Nourah bint Abdulrahman University Researchers Supporting Project: PNURSP2025R444.
Princess Nourah bint Abdulrahman University, Riyadh, Saudi Arabia.

### Competing Interests

The authors declare there are no competing interests.

### Author Contributions

- Sumathi Robert Shanmugam conceived and designed the experiments, performed the experiments, analyzed the data, prepared figures and/or tables, authored or reviewed drafts of the article, and approved the final draft.
- Amany Anwar Saeed Alabdullah conceived and designed the experiments, analyzed the data, prepared figures and/or tables, authored or reviewed drafts of the article, and approved the final draft.
- Maha Hanis Alenezi conceived and designed the experiments, prepared figures and/or tables, and approved the final draft.
- Shahad Khalid Aldughyshim conceived and designed the experiments, prepared figures and/or tables, and approved the final draft.
- Maryam Fahad Alnemer performed the experiments, authored or reviewed drafts of the article, and approved the final draft.
- Wedad Khalid Almutairi performed the experiments, authored or reviewed drafts of the article, and approved the final draft.
- Ghaida Saad Alhadyan performed the experiments, authored or reviewed drafts of the article, and approved the final draft.
- Rasha Zaid Albugomi analyzed the data, authored or reviewed drafts of the article, and approved the final draft.
- Fatimah Abdullah Alkhulayfi analyzed the data, authored or reviewed drafts of the article, and approved the final draft.

### Human Ethics

The following information was supplied relating to ethical approvals (i.e., approving body and any reference numbers):

The protocol for this study was approved by the Princess Nourah bint Abdulrahman University Institutional Review Board (approval number: 22-0110; March 3, 2022).

## Data Availability

The data is available at Figshare: Shanmugam, Sumathi Robert (2025). Pediatric pain management -Rawdata.xlsx. figshare. Dataset. https://doi.org/10.6084/m9.figshare.28278791.v1.

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
