# Peer review of "Knowledge and attitudes toward pediatric pain management among nursing interns from a selected university in Riyadh, Saudi Arabia: a descriptive cross-sectional quantitative study"

_PeerJ, doi:10.7717/peerj.19288_

## Round 0.1 · original submission · Major Revisions

Kindly revise your manuscript according to the reviewer's comments.

Please add more information about the study, especially regarding pediatric pain management.

Provide a revised version and a detailed response letter addressing each comment.

·

Basic reporting

Thank you for letting review this manuscript. Pediatriac pain management is of interest gloablly and I commend the authors for examining such topic.

Page 1, lines1-2: The title seems to be broad, yet it should be specific to indicate the context of the study. Hence the authors need to add the scope in terms of setting or at least the country. For example, they may consider formulating the title as “Knowledge and attitudes toward pediatric pain management among nursing interns from a selected academic institution in Riyad, Saudi Arabia”

Under section of Introduction:
Page 2, lines 45-46, consider adding a reference in support of the statement.
Line 49- throughout the manuscript. I have noted some inconsistency in referencing style for the in-text citations. Authors should consider using either the name and year of publication OR paraphrase to refine the statement and use numbering throughout the manuscript.

Experimental design

Under section of Materials and methods:
Page 4, lines 87-89; It is not clear about why selecting female nursing interns only. It is important the authors describe the study participants, if this category of nursing students are the only ones attending the academic institution, let the readers know about it.
Lines 92- 94; “The following EPI ap formula….”; authors need to add a reference.

Under section of instrument:
Page 5, line114-115; it not clear whether the experts review was initiated by the primary author (Manworren, 2000) or the authors of this study. Authors are suggested to add a statement clearly indicating that the tool was used as it was initially developed by the primary authors OR if there were some modifications/ any adaptation to the context.

Under section of Data collection:
Page 5, lines 117-118; the authors need to provide details on how the data collection procedure was done e.i., how was the participants’ recruitment done, how the participants accessed the google forms questionnaire,… For instance, the authors need to indicate whether they coordinated with program leaders or if liaised with interns’ representative to reach out to the participants, whether the forms were sent via emails, or other means as WhatsApp.

Under section of Statistical analysis:
Page 5 line 121; The authors need to be consistent in referencing style (numbering).
Line 123; it is written “ r value was used as a measure of association”, I did not see an information describing whether there was association or not among variables. No r values seen neither in the manuscript nor in the supplement files.
Given the demographic characteristics of study participants were collected, the expectation was that further statistical analysis than frequencies and percentages would be done to determine whether the levels of knowledge about and attitudes towards pediatric pain management could be associated with the demographics of participants.

Validity of the findings

Under section of Results:
Page 6, line 142 ; “ good level (75-100) of knowledge and positive attitude” - for a better understanding of the findings, I suggest that the authors indicate how the cut-off of 75 and above was determined for good level of knowledge and positive attitudes toward pediatric pain management.

Under section of discussion:
Page 7, lines 162-163, it is advisable to specify that the authors are referring to all the study participants, otherwise female participants were included in other studies such as the one conducted by Alotaibi (2019) in which female represented 39.1% of all the participants.
Page 8, line 173, consider adding the reference here to indicate the source of this information.
Lines 189-190; suggestion to refining the in-text citations for consistency with regard to reference style.

Additional comments

Thank you again for the opportunity to review your manuscript. Please, my comments are meant to be constructive, take them for what they are.

All the best.

Reviewer 2 ·

Basic reporting

A very interesting topic, especially for the training of future nurses.
The background is explicit, but it needs more substantiation, especially in the bibliography presented. I suggest that you present studies on the training of future nurses and the number of hours devoted to pain and pediatric pain in the basic course.

Experimental design

How did you get to 119 nurses? It is not clear to the reader how, out of 171 nurses, they obtained a sample of 119; this aspect needs further explanation.
It would also be important to mention in the ethical considerations that informed consent was entered into Google Forms. It should also be mentioned what measures were introduced to verify the veracity of the answers and that it was humans who responded
The reporting of the study followed the guidelines of the Strengthening the Reporting of Observational Studies in Epidemiology (STROBE) ? I don't see this mentioned, and it's methodologically essential

Validity of the findings

The results indicate a low level of knowledge regarding paediatric pain management. In the discussion, more reasons should be given for this than just a lack of practical experience. There is one aspect that they don't mention but which is curious: all of these nurse interns are affiliated with a single nursing school or several. Is the school curriculum common in Saudi Arabia or is it diverse? In short, the preparation of these nurse interns is a focus that needs to be addressed, because there are knowledge deficits that should be acquired in basic training, namely pharmacological ones.O

Additional comments

I continue to support the relevance of the study, but I think it should be strengthened with more bibliographic support.

---

## Round 0.2 · Minor Revisions

Minor revision is still required to accept your manuscript.

·

Basic reporting

no comment

Experimental design

no comment

Validity of the findings

no comment

Additional comments

Under the section of limitations, page 11, lines 218-220, in the pdf file it is written “The nursing curriculum’s coverage of pharmacological management was limited, as the course needed to provide an overview of all aspects of pediatric nursing care.” Is this authors’ perception? or they already know which pain management covered was content? if yes, was the case I would suggest including a statement in the last paragraph of the introduction indicating the issue. Nevertheless, they want to assess the level of knowledge and attitudes regarding ppm among the students.

---

## Round 0.3 · accepted · Accept

Thanks for improving your manuscript